# Synthesis, Cytotoxic Activity and In Silico Study of Novel Dihydropyridine Carboxylic Acids Derivatives

**DOI:** 10.3390/ijms242015414

**Published:** 2023-10-21

**Authors:** Ricardo Ballinas-Indilí, María Inés Nicolás-Vázquez, Joel Martínez, María Teresa Ramírez-Apan, Cecilio Álvarez-Toledano, Alfredo Toscano, Maricarmen Hernández-Rodríguez, Elvia Mera Jiménez, René Miranda Ruvalcaba

**Affiliations:** 1Departamento de Ciencias Químicas, Facultad de Estudios Superiores Cuautitlán Campo 1, Universidad Nacional Autónoma de México, Avenida 1o de Mayo s/n, Colonia Santa María las Torres, Cuautitlán Izcalli 54740, Mexico; ricardo_snf.49.ers@hotmail.com (R.B.-I.); nicovain@yahoo.com.mx (M.I.N.-V.); atlanta126@gmail.com (J.M.); 2Instituto de Química, Universidad Nacional Autónoma de México, Circuito Exterior s/n, Ciudad Universitaria, Mexico City 04510, Mexico; mtrapan@unam.mx (M.T.R.-A.); cecilio@unam.mx (C.Á.-T.); rtoscano@iquimica.unam.mx (A.T.); 3Laboratorio de Cultivo Celular, Sección de Posgrado e Investigación, Escuela Superior de Medicina, Instituto Politécnico Nacional, Mexico City 11340, Mexico; dra.hernandez.ipn@gmail.com (M.H.-R.); elviamj@gmail.com (E.M.J.)

**Keywords:** ynones, dihydropyridine carboxylic acid, cytotoxic activity, in silico DFT

## Abstract

To aid the possible prevention of multidrug resistance in tumors and cause lower toxicity, a set of sixteen novel dihydropyridine carboxylic acids derivatives **3a–p** were produced; thus, the activation of various ynones with triflic anhydride was performed, involving a nucleophilic addition of several *bis*(trimethylsilyl) ketene acetals, achieving good yields requiring easy workup. The target molecules were unequivocally characterized by common spectroscopic methods. In addition, two of the tested compounds (**3a**, and **3b**) were selected to perform in silico studies due to the highest cytotoxic activity towards the HCT-15 cell line (7.94 ± 1.6 μM and 9.24 ± 0.9 μM, respectively). Employing theoretical calculations with density functional theory (DFT) using the B3LYP/6-311++G(d,p) showed that the molecular parameters correlate adequately with the experimental results. In contrast, predictions employing Osiris Property Explorer showed that compounds **3a** and **3b** present physicochemical characteristics that would likely make it an orally active drug. Moreover, the performance of Docking studies with proteins related to the apoptosis pathway allowed a proposal of which compounds could interact with PARP-1 protein. Pondering the obtained results (synthesis, in silico, and cytotoxic activity) of the target compounds, they can be judged as suitable antineoplastic agent candidates.

## 1. Introduction

The group of α,β-acetylenic ketones, more commonly known as ynones, are molecules of great interest due to their importance as building blocks in the synthesis of many heterocycles; several of them are secondary metabolites, bearing biological activities and pharmaceutical applications [1,2,3,4,5,6]. This kind of compounds is highly reactive due to the presence of a carbonyl and an alkyne group [7,8]; it should also be noted that they are highly reactive Michael systems [9].

The pyridine nucleus is a versatile building block in organic synthesis [10,11] that is susceptible to activation by different agents; for example, it reacts with triflic anhydride (Tf_2_O) [12]. Furthermore, it is important to note that *bis*(trimethylsilyl) ketene acetals are nucleophiles that react with pyridinium salts supplying functionalized dihydropyridines [13,14,15]. Moreover, many pyridine derivatives such as 1,4-dihydropyridines are of special interest in drug design since they display interesting applications, for example: antibacterial [16], antifungal [17], anti-HIV [18], antihypertensive [19], and anticancer activities, among others, highlighting that any modification within the structure of dihydropyridine can be associated with excellent medicinal effects, decrease tumor growth, cell migration, and principally to avoid resistance of cancer cells against anticancer drugs [20]. As a part of our research interest, some members of our group have appropriately employed *bis*(trimethylsilyl) ketene acetals in combination with aza-aromatic substrates activated with triflic anhydride as an electrophilic activating agent to produce dihydropyridine carboxylic acids [21].

Cancer is a serious health problem, placed as the second leading cause of death worldwide. It is most manifested in the prostate, lungs, bronchi, uterine corpus, breasts, colon, liver, stomach, and blood [22]. The most frequently used cancer treatments are surgery, radiation, chemotherapy, hormonal therapy, and targeted therapy. However, many treatments are expensive, and in the treatment of malignant tumors, neoplastic cells can generate multidrug resistance (MDR) with different structures and functions. Additionally, current anticancer drugs are associated with several severe side effects [23]. Also, in anticancer drugs based on metal, the metal can bind to sites that should be occupied by other metals [24]. Consequently, the development of new molecules that avoid MDR in tumors, have a better pharmacological profile, and have lower toxicity for healthy cells is necessary. It is known that 1,4-dihydropyrine derivatives overcome MDR [25]. However, their clinical use is limited due to their strong vasodilator activity, and new chemotherapeutics that do not present this drawback are needed to overcome MDR in cancer treatment [26].

On the other hand, DFT calculations are favored nowadays for replications of experimental facts such as molecular geometry, vibrational frequencies, and chemical shifts, including other properties of organic molecules [27]. In this sense, calculated frequencies and chemical shifts are scaled to compensate for the estimated handling of electron correlation, basis set deficiencies, and anharmonicity [28,29], creating good approximations for organic molecules [27,30].

Considering the commentary above, the goal of this work involves the use of several *bis*(trimethylsilyl) ketene acetals to produce sixteen novel carboxylic acid derivatives and their corresponding cytotoxic evaluation in several cancer cell lines (U251, PC-3, K-562, HCT-15, MCF-7, and SKLU-1) and normal cell (COS7). Complementary, physicochemical, and toxicological predictions; in silico studies (docking) to predict the affinity to proteins related to apoptosis induction; and quantum chemistry for the determination of molecular properties were performed.

## 2. Results and Discussion

### 2.1. Synthesis

Dihydropyridine carboxylic acid **3a** was produced (Figure 1) by fine-tuning a procedure from previous research [31]: the ynone **1a** was activated with a slight excess (1.2 equiv) of triflic anhydride for 3h, followed by a nucleophilic addition (at C6) of 1.2 equiv of the *bis*(trimethylsilyl) ketene acetal (**2a**) in 15 mL of anhydrous CH_2_Cl_2_ (under an inert atmosphere at −78 °C), and stirring continued at −78 °C for another 8 h. To generalize the procedure (Table 1), a set of sixteen (**3a–p**) new molecules were synthesized, using different ynones as substrates, in the presence of several *bis*(trimethylsilyl) ketene acetals. In general, the reaction yields ranged from regular to good, even when comparing the reaction with imidazoles as substrates as observed in previous research [31]. The procedure resulted in easy workup, achieving sixteen new products; it will be demonstrated that several of the products are good antineoplasic candidates.

For the structural attribution of the obtained molecules, the target **3a** is offered as an example: it is a white solid and obtained with a 91% yield, displaying an mp of 122–124 °C; implying a pure compound, the infrared spectrophotometric data (cm^−1^) showed a broad band centered at 3086 cm^−1^, which was assigned to the OH of the carboxylic group; a sharp medium band was observed at 2199 cm^−1^, which unequivocally belongs to the C≡C system; as expected, two intense bands were present at 1705 and 1625 cm^−1^, which were assignable to the C=O group and to the system C=C, respectively. The corresponding ^1^H NMR spectrum displayed a broad singlet at 11.28 ppm, which was unequivocally assigned to the proton of the carboxylic moiety; a simple signal at 8.18 ppm was assigned to H5, denoting a displacement to high frequency due to the electron-withdrawing influence of the α,β-unsaturated ketone system; at 6.77, 5.55, and 4.16 ppm, a double, a double of double and another double signals were presented. These signals are assigned, respectively, to H9, H8, and H7, which is in agreement with the corresponding coupling data (*J_9–__8_* = 5.4 Hz, *J_8–7_* = 1.8 Hz); a double and multiple signals were observed in the range of 7.63–7.45 ppm, which is conveniently assigned to the phenyl group. Related to the ^13^C NMR spectrum, the main signals that agree appropriately with the **3a** structure were assigned to the carbonyl groups at 182.3 and 176.1 ppm of the carboxylic moiety and ynone, respectively. Likewise, the carbons corresponding to the triple bond are highlighted at 92.5 and 84.6 ppm. Associated to the HRMS, acquired by the DART^+^ mode (19 eV), an elemental composition C_19_H_17_F_3_NO_5_S was stated, in agreement with a fragment ion [M+1]^+^; this fragment was unequivocally correlated (error of −1.37 ppm) with an exact value of 428.0779 Daltons and a precise value of 428.0773 Daltons, complementarily the provided unsaturation data (11.5) was in agreement with the structure. Additionally, it is important to comment that the spectroscopic spectra for all compounds, Appendix A, is offered in Appendix A.

It is convenient to highlight that the molecular structure of the title molecules was unequivocally confirmed, correlated through an X-ray diffraction analysis; molecule **3f** is shown as an example in Figure 1.

In this sense, suitable X-ray quality crystals of **3f** were grown by slow evaporation from an *n*-hexane/CH_2_Cl_2_ solution at 4 °C. Details of the data collected and the structure refinement parameters employed are summarized in Table 2. The dihydropyridine carboxylic **3f** crystallizes in monoclinic (space group P21/c) with the cell measurements of a = 5.8706(1) Å, b = 15.1627(2) Å, c = 20.9890(3) Å, and α = 90°, β = 93.576°, and γ = 90°, with volume of 1864.68 Å^3^ and Z of 4 at 25 °C. The complete refined crystal structure specification using SHELXS-2013 is summarized in Table 2.

### 2.2. Cytotoxic Assay

The compounds were tested in vitro for cytotoxic activity (Table 3) and were determined by using the protein-binding dye at 25 μM for **3a**–**p** in six tumor cells: U251 (human glioblastoma), PC-3 (human prostatic adenocarcinoma), K-562 (human chronic myelogenous leukemia), HCT- 15 (human colorectal adenocarcinoma), MCF-7 (human mammary adenocarcinoma), and SKLU-1 (human lung adenocarcinoma) were determined by using the protein-binding dye sulforhodamine B (SRB) assay in microculture to determine cell growth [32].

The initial cytotoxicity screening data listed in Table 3 show moderate inhibition to tumor cell lines. However, only nine compounds (**3a**, **3d**, **3e**, **3i**, **3j**, **3m**, **3n**, **3o**, and **3p**) showed an inhibition percentage greater than 50% in the HCT-15 cell line; however, it is important to mention that several of these compounds display cytotoxic effect against normal cells (COS7); consequently, these compounds were discarded for further studies. Also, it is important to mention that the compound **3b** was used to further studies due to displaying an inhibition of 42.7%, but does not show cytotoxicity against COS7. Finally, we determined the IC_50_ of the most active compounds over HCT-15 (Table 4).

Analyzing these results, we observe that only the compounds **3a** and **3b** display the lower IC_50_ values in comparison with both references Cisplatin and Gefitinib, which indicates their good activity against the evaluated tumor cell line.

### 2.3. In Silico Evaluation of Physicochemical Properties

According to the above results, the physicochemical characteristics of compound **3a** and **3b** were analyzed by employing the Osiris Property Explorer. As can be seen in Table 5, both compounds show physicochemical properties that are in accordance with Lipinski’s rule of five (molecular weight < 500 g/mol, Log *p* < 5, hydrogen bond donator < 5, and hydrogen bond acceptor < 10). This rule describes molecular properties important for a drug’s pharmacokinetics in the human body, including absorption, distribution, and excretion (named ADME), as these physicochemical properties may represent the difference between failure and the development of a successful oral drug [33].

In addition, the topological polar surface area (TPSA), which is defined as the surface total over all polar atoms, including oxygen, nitrogen, etc., and their bonding hydrogen atoms, were calculated for compounds **3a** and **3b**. TPSA is a commonly used medicinal chemistry metric for the optimization of a drug’s ability to permeate cells [34]. Molecules with a TPSA value higher than 140 angstroms squared (Å^2^) tend to be poor at permeating cell membranes. As shown in Table 5, the TPSA values for studied compounds are lower than 140 Å^2^.

The other biological properties evaluated were mutagenicity, teratogenicity, irritant and reproductive effects to computational assessment of studied compounds. Only compound **3a** showed low risk for reproductive effects, while compound **3b** was shown to be safe according to the computational evaluation. Therefore, all estimates indicate that there is a high possibility that the designed molecules should be able to pass through lipid membranes when they are experimentally evaluated in animal or human models [35].

### 2.4. Docking Studies

Docking studies represent a computational tool that are highly employed to propose appropriately, at an atomic level, interesting protein–ligand interactions [36]. It is worth noting that numerous docking studies with proteins involved in apoptosis pathway have been described elsewhere [37]. The averages of ΔG values obtained by docking studies with proteins related to apoptosis cascade and cell cycle regulators are summarized in Table 6. As it can be seen, for proteins related to apoptosis regulation, compounds **3a** and **3b** exhibit more affinity to PARP-1 protein than the rest of the studied proteins, evidenced by the lowest values of ΔG (−9.8 and −10.6 kcal/mol, respectively). The binding mode and the interactions established with PARP-1 in the complex with the lowest affinity and the highest frequency are shown in Figure 2 and described below.

PARP-1 is an enzyme which plays a pivotal role in cellular biological processes, including DNA repair. Interestingly, PARP-1 has been found to be overexpressed in various carcinomas, including colorectal cancer [39]. These findings highlight the clinical potential of PARP-1 inhibitors as a therapeutic target of human cancer. Additionally, multiple preclinical trials demonstrate that the inhibition of PARP-1 can repress tumor growth and metastasis [40]. As shown in Figure 2a, compounds **3a** and **3b** bind to the catalytic site of PARP-1. Figure 2b shows the interaction of a known inhibitor (FR257517) with PARP-1 by stablishing three hydrogen bonds with Ser243, Ser203, and Gly202; a π–π interaction with Tyr246; in addition to a set of van der Waals contacts with Arg217, Leu108, Asp109, Asp105, Asn207, Gly202, Phe236, Ala 237, and Tyr235, as was reported previously [38]. Likewise, the analysis of the conformation with the lowest ΔG and the highest frequency obtained by docking studies showed that the studied compound interacts with PARP-1 protein through a π–π interaction with Tyr246; and through hydrogen bonds with Asp105, Arg217, Asp109, and Ser203; in addition to a set of van der Waals contacts with Asn106, Asn207, Tyr235, Hys201, Gly202, Glu327, Ala237, and Lys242, Figure 2c. In general, the high affinity of studied compounds to PARP-1 could be explained by two important characteristics: a small pocket, where compounds **3a** and **3b** can fit during the interaction; and the catalytic site of the protein is formed by polar amino acids allowing it to target compounds to establish hydrogen bond interactions that stabilize the binding mode. Thus, the interaction of compounds **3a** and **3b** could explain their cytotoxic effect in the HTC-15 cell line.

### 2.5. Quantum Chemistry Studies

Taking into account the aforementioned comments, the molecular properties and the experimental characterization of **3a** and **3b** were contrasted with their corresponding theoretical data obtained from DFT calculations, using the B3LYP with 6-311++G(d,p) basis set. The theoretical calculations were related to the chemical shifts, in ppm, with nuclear magnetic resonance (NMR); for the vibrational modes, they were related, in cm^−1^, with infrared spectrophotometry; these analyses were performed for the most stable conformer of **3a** and **3b** molecules in the gas phase.

#### 2.5.1. Geometrical Parameters for **3a** and **3b**

The most stable structures of **3a** and **3b**, Figure 3, were computed with optimized energies of −116.867 × 10^3^ kcal/mol and −119.257 × 10^3^ kcal/mol. Selected geometric parameters (bond lengths, bond angles, dihedral angles) of compounds **3a** and **3b** are shown for comparative analysis with XRD-**3f** values in Table 7. Most of the experimental values fit with DFT data calculated in the gas phase while XRD-**3f** is in a solid phase. The sp^3^, sp^2^, and sp carbon–carbon bond lengths in **3a** and **3b** range from 1.198 to 1.573 Å (XRD-**3f**) and from 1.211 to 1.597 Å (B3LYP). The carbon–oxygen bond lengths for **3a** and **3b** range from 1.218 to 1.317 Å (XRD-**3f**) and from 1.207 to 1.355 Å (B3LYP). Some lengths showed variations between 0.003 and 0.119 Å, for molecules **3a** and **3b** with respect to XRD-**3f** [41]. The C2–C7 and C1–C2 bond lengths of the **3a** molecule showed deviations of 0.021 Å and 0.012 Å with respect to **3b**, respectively. Moreover, the bond angles of C7–C2–C1 of **3b** showed a slightly increased deviation, 2.3°, with respect to **3a**. Some bond angles showed variations between 0.01 and 4.9° with respect to XRD-**3f**. The dihedral angles in Table 5 for **3a** and **3b** are similar; the variation between them is 0.5 and 6.6°. The type of substituent at N3 does not contribute to the planarity of a certain part of the molecules **3a** and **3b**. Therefore, the C5–C6–C7–C2, C9–C8–C7–C2, and C11–C6–C7–C2 dihedral angles showed some differences between them, 5.1, 5.9, and 6.6°, respectively.

#### 2.5.2. Frontier Molecular Orbital (FMO)-Reactivity Parameters of **3a** and **3b**

The electron that donates the outermost molecular orbital packed with electrons is identified as the highest occupied molecular orbital (HOMO), while the electron that accepts the lowest empty molecular orbital is the lowest unoccupied molecular orbital (LUMO). The DFT approach reveals that molecules **3a** and **3b** have 752 and 774 molecular orbitals including 110 and 113 orbitals filled by electrons and 642 and 661 empty orbitals, respectively. In the present computational study of the title of compounds **3a** and **3b** by the DFT method, the representation of molecular orbitals HOMO (orbital number: 110 and 113) and LUMO (orbital number: 111 and 114) with their corresponding energies (in eV) are illustrated in Figure 4. The energies of HOMO and LUMO, for **3a** and **3b**, are −6.806 eV and −6.718, and −2.602 and −2–622 eV, respectively.

The difference in energy between HOMO and LUMO (frontier molecular orbital analysis) can be employed to predict the bioactive potency (reactivity) and chemical stability of the molecule [42,43]. In this sense, quantum molecular reactivity parameters like electronegativity (χ), chemical potential (μ), electrophilicity index (ω), global hardness (η), and charge transfer capability (Δ) were calculated by HOMO and LUMO energies [44] for compounds **3a** and **3b**, Table 8.

Accordingly, the ionization energy of the compounds changed as **3a** (8.121 eV) > **3b** (8.053 eV), and the electron affinity changed as **3b** (1.217 eV) > 3a (1.190 eV) in the gas phase. As far as the energy gap values, it can be observed that intermolecular interactions for **3b** are more likely than in **3a** because the ΔE_LUMO-HOMO_ for the compounds is calculated as **3a** (4.204 eV) > **3b** (4.096 eV); it can be said that compound **3a** is harder (3.466 eV) than compound **3b** for the gas phase. In this study, the electrophilicity index values of the compounds support the relative acidic characterization. In Table 6, compound **3b** (3.142 eV) has a more electrophilic character than compound **3a** (3.127 eV). Furthermore, compound **3b** (0.678 eV) has a slightly higher capability of charge transfer than compound **3a** (0.672 eV).

It is well known [45,46,47] that the ΔE_LUMO-HOMO_ has been normally used to provide evidence on the kinetic stability and reactivity of related molecular systems. Thus, the nucleophilicity of compound **3a** is greater than compound **3b**, which can be the reason for the anticancer activity of compound **3a**. The highest value of chemical potential occurs in **3a** (−4.656 eV). This property may be related to the nucleophilic character of the molecule being the most reactive. The highest value of electronegativity is for **3a** (4.656 eV), which indicates that **3a** is capable of attracting electrons to itself.

Occupied orbitals (HOMO) of **3a** and **3b** have contributions of the oxygen of hydroxyl and carbonyl groups, triple bond and phenyl ring. These moieties could be considered to be nucleophilic sites that could account for the antiproliferative activity of **3a** and **3b** against the studied human cancer cells. The carbon of carbonyl group and phenyl ring contributed to virtual frontier molecular orbitals, pointing to them as potential electrophilic sites for **3a** and **3b** and derivatives’ antiproliferative activity against cancer cells.

#### 2.5.3. Molecular Electrostatic Potential (MEP) of **3a** and **3b**

The MEP plots indicate the electrophilic and nucleophilic sites. The positive potential is definite with a blue color, which establishes the electrophilic attack center, while the negative potential is visualized in red, representing the nucleophilic attack center. In Figure 5, the MEP plot for **3a** and **3b** shows that the negative potential is mostly due to lone pairs of the oxygen of the carbonyl group and the hydroxyl oxygen; the strongest is that of the oxygen of **3b**; its most negative value is −56.79 kcal/mol, and positive potential regions are mainly over the hydrogen atom belonging to the hydroxyl group.

Accordingly, molecules **3a** and **3b** could interact with a receptor site directly, and this interaction stabilizes the oxygen atoms of the carbonyl group of these molecules. The large positive potential in the hydrogen atom may have a contributing effect toward the association of these compounds with the receptor site, as it is well known that positively charged hydroxyl hydrogen atoms have a high affinity for negative centers.

#### 2.5.4. Atomic Charges for **3a** and **3b**

The atomic charges in a compound are a key feature in numerous applications of quantum mechanical calculations to evaluate various properties of molecules [48,49]. By analyzing charge distributions in molecules, details about donor and acceptor pair formation can be obtained, which involves charge transport inside the molecule. The full atomic charges for both compounds are given in Table 9. Here, the natural atomic charge calculations revealed that the positive charge was located on the electropositive H and S atoms as expected; the net atomic charges for the O atoms of compounds **3a** and **3b** were calculated as −0.574 to −0.870 e-. However, the net charge for the O atom of sulfate groups was predicted between −0.860 e- and −0.870 e-. The net charges of the H atoms in compounds **3a** and **3b** were calculated as +0.224 to +0.487e- and +0.223 e- to +0.480, respectively. Furthermore, the charge distribution over the triple bond, carbonyl group, and C14 phenyl moiety of both compounds showed a change due to the resonance effect. The sulfur atom exhibits a positive charge due to the bonds with the oxygen and nitrogen atoms. Also, the C21 atom through the bonds with the fluorine atoms.

As seen from the MEP surfaces and the charge analysis, hydrogen in the hydroxyl group for **3a** and **3b** is acidic (Lewis’s acid); and the carbonyl in the ketonic and carboxylic groups is more electronegative (Lewis’s base).

Consequently, the spin density derived from NPA calculations can be used to explain the electronic distribution in the molecular systems. The most positive or most negative NPA charges are indicators of the distribution of the electronic density of atoms for **3a** and **3b** molecules; and the availability of attracting or donating electrons atoms are indicators for the formation of intermolecular interactions, hydrogen bonds, π–π interaction, and van der Waals contacts between these molecules and amino acids residue of protein.

#### 2.5.5. Theoretical ^1^H and ^13^C NMR Spectra of **3a** and **3b**

The NMR theoretical chemical shift for ^1^H and ^13^C in the gas phase for **3a** and **3b** were calculated using B3LYP with the 6-311++G(d,p) basis set and the GIAO method. Hence, Figure 6 and Figure 7 display the correlation employing linear regression, and Table 10 displays both ^1^H and ^13^C NMR theoretical and experimental chemical shift values. Related to the ^1^H NMR spectra for both molecules (**3a**, **3b**), it is important to mention that the signal of OH influences the linearity, Figure 6a and Figure 7a. Therefore, the theoretical values for this signal show a chemical shift of 5.7737 ppm, and the experimental results presented a value of 11.28 ppm for **3a**; furthermore, for **3b**, the corresponding values were 5.7251 and 9.87 ppm for the theoretical and the experimental chemical shift, respectively. These differences may be explained considering that in the experimental spectrum, the OH group displays a common dissociation, producing an intramolecular hydrogen bond with the oxygen of the carbonylic group. This could not succeed in the gas phase. Thus, these hydrogen bonds could modify the electron density, consequently diminishing the regression coefficient value [50]. Additionally, the small discrepancy may be because the experimental NMR was developed in the liquid phase, and theoretical calculation was conducted in the gas phase [51,52].

On the other hand, removing this signal from the correlation, Figure 6b and Figure 7b, excellent regression coefficient values were obtained, R^2^ = 0.9984 with a corresponding standard deviation of 2.5803 and a typical error of 0.1135 ppm for **3a**. In this sense, the equation to describe the fit is δCalc = 1.036δexperimental–0.105 ppm. The slope and intercept values are 1.04 and 0.11 ppm, respectively. For **3b**, the value is R^2^ = 0.9948 with a standard deviation of 2.3787 and a typical error of 0.1847 ppm; the equation to describe the fit is δCalc = 1.0093δexperimental + 0.0608 ppm. The slope and intercept values are 1.00 and 0.06 ppm, respectively.

On behalf of the ^13^C NMR, Figure 6c and Figure 7c, the linear regression results show a regression coefficient of 0.9971 and 0.9980 with a standard deviations of 47.4734 and 48.2897, with typical errors of 2.66 and 2.19 ppm for **3a** and **3b**, respectively: the equation to describe the fit for **3a** is δCalc = 1.011δexperimental + 4.732 ppm, with 1.01 and 4.73 ppm being the slope and intercept, respectively; for **3b**, the equation is δCalc = 0.9788δexperimental + 9.4635 ppm, with a slope of 0.98 and an intercept of 9.46 ppm.

These results showed a good prediction of the chemical shifts, considering the regression value, indicating a good agreement between theoretical and experimental values.

#### 2.5.6. Theoretical Infrared Spectra of **3a** and **3b**

The side chain on nitrogen atom does not have any important contribution to the dihydropyridine ring; therefore, this work was focused on the ring, the carbonylic groups, and the alkyne substituents. In this sense, the analyses are focused on specific vibrational modes: C5=C6, C8=C9, C7-H, C11=O, C1=O, C1-OH, and C12≡C13 for molecule **3a**, and C3=C4, C6=C7, C5-H, C12=O, C1=O, C1-OH, and C13≡C14 for molecule **3b**, Table 11. The **3a** and **3b** molecules have 45 and 46 atoms, exhibiting 129 and 132 normal active vibrational modes, respectively.

##### Experimental and Theoretical Vibrational Spectra

The experimental spectrum of **3a** and **3b**, Figure 8, showed a weak broad band at 3086 and 3069 cm^−1^ assigned to the OH of the carboxylic group, generally found as a dimer. These values agree with the reported range of 3300 to 2500 cm^−1^, usually centered at 3000 cm^−1^ [53]. Finally, the weak band is probably due to the intramolecular hydrogen bond with a carbonyl group (ketone) [54]. Additionally, the C=O stretching intense band appearing at 1705 and 1611 cm^−1^ and 1703 and 1640 cm^−1^ are assigned to the carboxylic acid and ketone groups for **3a** and **3b**, respectively, which is consistent with the literature value, 1870–1540 cm^−1^ [53,55]. The 2199 cm^−1^ value which appears corresponds to alkyne group as a medium band is congruent with the range previously reported, 2260–2100 cm^−1^ [53]. Regarding the 1,4-dihydropyridine system, two stretching medium bands were found for C=C groups at 1671 and 1625 cm^−1^ and 1678 and 1640 cm^−1^ for **3a** and **3b**, respectively. These results are consistent to those reported by Soliman et al. [56] at 1600 cm^−1^ for dipyrazolopyrimidine-1,4-dihydropyridines and Tu et al. [57] at 1652 to 1604 cm^−1^ for dihydropyridines. Finally, a weak stretching band at 2942.64 and 2935 cm^−1^ was observed for C7–H and C5–H bonds, respectively, and agrees with the literature data [53,58].

Regarding the theoretical level, Figure 8 and Table 11, it was observed that the OH for acidic moiety displays a weak band at 3643 and 3632 cm^−1^ for **3a** and **3b**, respectively; these bands probably recognize the OH of carboxylic acid as a free hydroxyl. These values are according to those reported by Devi et al. [59] at 3634 cm^−1^ using the B3LYP method with a 6-31G(d,p) basis set. Also, Devi et al. [60] reported the value for OH at 3545 cm^−1^.

The bands C7–H and C5–H at 2942 and 2935 cm^−1^ are presented as weak bands for **3a** and **3b**, respectively. In this case, Karthick et al. [61] detected values in the span of 2979–2978 cm^−1^ for 1,4-dihydropyridines, Kavitha and Verlaj [62] detected values between 3144 and 2900 cm^−1^ for this band for isoxazoles; Radder et al. [51] reported the value at 3029 cm^−1^ for a benzonitrile derivative; and Fekri and Nikpassand [54] reported values comprising 3077–2987 cm^−1^ for dihydropryridine derivatives.

Related to C=O bands, two carbonyl groups are presented: C=O for the carbonyl of the acid moiety, which is displayed at 1738 cm^−1^ for **3a** and 1730 cm^−1^ for **3b** as medium bands. In this sense, Devi et al. [59] detected this band at 1766 cm^−1^; Devi et al. [60] reported this band at 1729 cm^−1^. For C=O, the carbonyl of the ketone moiety is displayed as medium bands at 1621 cm^−1^ for **3a** and 1617 cm^−1^ for **3b**; this band is easily to identify due to higher strength and conjugation with the double bonds C5=C6 or C3=C4. Karthick et al. [61] report this band at 1697–1685 cm^−1^; Fekri and Nikpassand [54] report it at 1657–1643 cm^−1^ for 1,4-dihydropyridines and at 1652 cm^−1^ for a benzonitrile derivative [51].

At 1596 and 1656 cm^−1^ for **3a** and 1596 and 1663 cm^−1^ for **3b**, weak bands for C8=C9 and C6=C7 bonds were detected and medium bands for C5=C6 and C3=C4 bonds of the dihydropyridine ring were detected, respectively; probably, these bands are displayed as a medium band due to conjugation with the carbonyl of ketone group. Kavitha and Verlaj [62] detected theoretical values from 1601 to 1491 cm^−1^, obtained for isoxazoles; Radder et al. [51] also reported values ranging from 1652 to 1577 cm^−1^ for a benzonitrile derivative, and 1499–1643 cm^−1^ for dihydropryridine derivatives [54].

Finally, the alkyne group C12≡C13 and C13≡C14 is detected at 2217 cm^−1^ for both **3a** and **3b** as intense bands, agreeing with the previously reported values of 2370 cm^−1^ for propargyl arms containing a Schiff base [63] and 1800 cm^−1^ for complex between terminal alkynes with ruthenium [64].

These results demonstrate a good correlation between the experimental and theoretical data, correctly allowing for band assignments.

## 3. Materials and Methods

### 3.1. General

All reagents and solvents were of analytical grade, acquired from commercial suppliers, and used without further purification. Melting points were measured on a Melt Temp II apparatus and are uncorrected. The IR spectra were recorded on a Bruker TENSOR 27 spectrophotometer. The ^1^H and ^13^C NMR spectra were performed on a Bruker Advance III apparatus at 300 MHz and 75 MHz, respectively, in chloroform-*d*; the corresponding δ-chemical shifts are given in ppm, utilizing TMS as the internal reference. HRMS-DART^+^ (19 eV) spectra were obtained with a JEOL JMS-T100LC spectrometer.

Suitable crystals of **3f** were grown in a *n*-hexane/CH_2_Cl_2_ system at 4 °C for more detailed scrutiny of its molecular structure with X-ray diffraction. The crystal was placed on a glass fiber at 25 °C and then put in a Bruker Smart Apex CC diffractometer equipped with Mo radiation (λ_MoKα_ 1⁄4 0.71073 Å). The decay was negligible in all cases. Systematic absences and intensity statistics were employed for space group determination. The structure was established with direct methods on the SHELXS-2013 program. The solution and refinement of structures were performed with SHELXL-2013. The refinements of the anisotropic structure were made with the least squares technique for all non-hydrogen atoms. The hydrogens were placed in idealized positions based on their hybridization with isotropic thermal parameters fixed at 1.2 times the value of the attached atom. The different acetals of *bis*(trimethyl) silyl ketene **2a**–**d** were prepared according to methods reported in the literature [65]. The substrate ynones (**1a**–**h**) were prepared accordingly to a reported approach [66].

### 3.2. Ab Initio Calculations

Mechanic molecular [67,68,69] geometries of a conformational analysis previous, using the Spartan06 (Version 06, Wavefunction, Inc., Irvine CA, USA) [70] program, were used as initial models for optimizations for the density functional theory (DFT) [71,72,73]. The geometry of the title compound **3a** and **3b** was optimized by functional B3LYP [74,75] and 6-311++G(d,p) method [76,77,78] using Gaussian 16 (Version 2016.03, Gaussian, Inc., 340 Quinnipiac St., Bldg. 40, Wallingford CT 06492, UK) [79] software and visualized by GaussView [80] application. The geometries found by theoretical calculations were considered as true energy minima only when all calculated frequencies (PES) were positive. Theoretical vibrational analysis of **3a** and **3b** were executed with the same theory approach, and computed frequencies were assigned to respective modes of vibrations using a scale factor of 0.9679 [27,30,81,82]. Optimized geometries were also used to reproduce ^1^H and ^13^C NMR chemical shifts obtained using the Gauge-Independent Atomic Orbital (GIAO) approach [83,84] with a global scaling factor of 31.9699 and 184.1279 for ^1^H and ^13^C, respectively, with respect to TMS in gas phase. Atomic charge distribution analysis for **3a** and **3b** was carried out by Natural population analysis (NPA) [85], by Natural Bond Orbital (NBO) analysis [86] implemented in Gaussian 16 software. Moreover, a frontier molecular orbital (FMO) [87] determination and global reactivity descriptors [42,88] were found out using HOMO and LUMO energy values, and molecular electrostatic potential (MEP) [89] diagrams were drawn to estimate/evaluate the chemical reactive behavior/site of two compounds.

### 3.3. In Silico Evaluation of Physicochemical Properties

The OSIRIS property explorer (http://www.organic-chemistry.org, accessed on 3 August 2023) was employed to predict the physicochemical properties of the compounds according to Lipinski’s rules [33] to select ones with characteristics favoring absorption and distribution after oral administration.

### 3.4. Docking Studies

To determine the interaction between selected compounds with proteins related to apoptosis induction, the 3-dimensional (3D) structure of the following proteins was retrieved from protein data bank: B-cell CLL/lymphoma 2 (Bcl-2) PDB ID: 4IEH, B-cell lymphoma-extra-large (Bcl-XL) PDB ID: 4QVF, Mouse double minute 2 homolog (MDM2) PDB ID: 4HG7, Mammalian target of rapamycin (mTOR) PDB ID: 4JT5, Histone deacetylase 8 (HDAC-8) PDB ID: 5FCW, and Poly (ADP-ribose) polymerase 1 (PARP-1) PDB ID: 1UK0. The water molecules, co-crystalized ligands, and peptides were eliminated, conserving only the cofactors necessary to the enzymatic activity for each protein.

Autodock 4.2 software was used for docking studies (http://autodock.scripps.edu, accessed on 18 October 2023) [90]. This program has been widely used because it displays good free energy and binding pose correlation values between docking simulations and experimental data [91]. Three-dimensional structures of compound **3a** and **3b** were generated as previously described. In addition, to validate the docking protocol, known inhibitors for the studied proteins were included in the docking studies, Table 12. A GRID-based procedure was utilized to prepare the structural inputs and define all the binding sites. A rectangular lattice (70 × 70 × 70 Å) with points separated by 0.375 Å was centered on the active site of each protein, Table 12. All docking simulations were conducted using the hybrid Lamarckian genetic algorithm with an initial population of 100 randomly placed individuals and a maximum of 1.0 x 10^7^ energy evaluations. All other parameters were maintained at their default settings. The resulting docked orientation within an RMSD of 0.5 Å was clustered together. The lowest energy cluster for each ligand was subjected to further free energy and binding geometry analysis, as previously reported [92]. Conformations with the lowest free energy binding (ΔG) and the highest frequency were selected employing AutoDock tools. The images were created by employing PyMol [93].

### 3.5. Cytotoxic Activity

The in vitro cytotoxicity of the compounds was evaluated on HCT-15 cell line (human colorectal adenocarcinoma), using the protein binding dye sulforhodamine B (SRB) in a microculture assay to measure cell growth [99]. The cell lines were grown in RPMI-1640 medium (Sigma Chemical Co., Ltd., St. Louis, MO, USA) supplemented with 10% fetal bovine serum (Invitrogen Corporation, Waltham, MA, USA), 2 mM L-glutamine, 10,000 units/mL of penicillin G, 10,000 μg/mL streptomycin, and 0.25 μg/mL of fungizone (Gibco, Amarillo, TX, USA). The cells were maintained at 37 °C in a 5% CO_2_ atmosphere with 95% humidity.

Preparations were made with 7.5104 cell/mL of the HCT-15 cell line, adding 100 mL of each of these suspensions to the wells of 96-well plates. Incubation was carried out for 24 h to promote cell attachment, and then, 100 μL per well of a test compound, gefitinib or cisplatin (as controls), was added. After 48 h, the adhered cell cultures were fixed in situ with a cold 50% (*w/v*) solution of trichloroacetic acid, followed by incubation at 4 °C for 1 h. Upon completion of this time, the supernatant was discarded, and the plates were washed and air dried. Cells cultured with TCA were stained with 100 μL of 0.4% SRB solution for 30 min. The protein-bound dye was extracted with a 10 nm unbuffered tris base, and the optical density was read at 515 nm on a Synergy HT Microplate Reader (Elx 808, BIO-TEK Instruments, Inc, Winooski, VT, USA). IC_50_ values were determined with the Monks protocol. Accordingly, a concentration–response curve was constructed for the dihydropyridine carboxylic acids **3a**, **3b**, **3f**, **3i**, **3o**, **3p**, and the values corresponding to inhibition of 50% (IC_50_) were calculated from non-linear regression equations. IC_50_ values are expressed as the mean ± standard error (SE) [100,101].

### 3.6. General Procedure for the Synthesis of Dihydropyridine Carboxylic Acids ***3a–p***

To a solution with 0.2 g (1.0 mmol) ynone derivative of pyridine dissolved in 15 mL of anhydrous CH_2_Cl_2_ (under an inert atmosphere at −78 °C), 0.19 mL (1.2 mmol) of triflic anhydride was added with a syringe, the mixture was stirred constantly during 3 h. Subsequently, 1.2 mmol of the corresponding ketene acetal was added and stirring continued at −78 °C for another 8 h. The reaction was then allowed to reach 25 °C before being transferred to a separatory funnel and washed with water (3 × 30 mL). The organic layers were combined and dried up with anhydrous Na_2_SO_4_; then, the solvent was removed by vacuum evaporation. Finally, the reaction crude was purified by column chromatography using silica gel 60 (0.063–0.200 mm, 70–230 mesh ASTM, acquired from Merck-millipore, Germany) with different *n*-hexane/ethyl acetate mixtures as eluents, obtaining the title pure compounds **3a–p**.

#### Spectroscopical Characterization of Dihydropyridine Carboxylic Acids **3a–p**

2-methyl-2-(3-(3-phenylpropioloyl)-1-((trifluoromethyl)sulfonyl)-1,4-dihydropyridin-4-yl)-propanoic acid (**3a**), white solid was obtained with 91% isolated yield (0.37g); mp 122–124 °C; IR (cm^−1^): 3086 (COOH), 2199 (C≡C), 1705 (C=O), 1625 (C=C); ^1^H NMR (CDCl_3_) δ (ppm): 11.28 (s, 1H, OH), 8.20 (s, 1H, H5), 7.63 (d, *J* = 6.6 Hz, 2H, H15, H19), 7.49 (m, 1H, H17), 7.45 (d, *J* = 7.5 Hz, 2H, H16, H18), 6.77 (d, *J* = 5.4 Hz, 1H, H9), 5.55 (dd, *J* = 1.8 Hz, *J* = 5.4 Hz, 1H, H8), 4.16 (d, *J* = 5.4 Hz, 1H, H7), 1.17 (s, 3H, H10), 1.16 (s, 3H, H3); ^13^C NMR (CDCl_3_) δ (ppm): 182.4 (C1, C=O), 176.1 (C11, C=O), 137.5 (C5), 133.5 (C15, C19), 131.0 (C17), 128.8 (C16, C18), 122.7 (C9), 122.9 (C6), 121.3 (C14), 119.1 (q, *J*_C-F_ = 321 Hz, C21, CF_3_), 112.2 (C8), 92.5 (C13), 84.6 (C12), 47.8 (C2), 37.5 (C7), 22.6 (C10), 19.8 (C3); HRMS-DART^+^ (19 eV): elemental composition C_19_H_17_F_3_NO_5_S, [M+1]^+^, correlation (error of −1.37 ppm) exact value 428.0779 Daltons and precise value of 428.0773 Daltons; complementarily, the provided unsaturation data (11.5) were in agreement with the structure.

1-(3-(3-phenylpropioloyl)-1-((trifluoromethyl)sulfonyl)-1,4-dihydropyridin-4-yl)-cyclobutane-1-carboxylic acid (**3b**), white solid was obtained with 82% isolated yield (0.34 g); mp 134–136 °C; IR (cm^−1^): 3069 (COOH), 2199 (C≡C), 1703 (C=O), 1640 (C=C); ^1^H NMR (CDCl3) δ(ppm): 9.87 (s, 1H, OH), 8.18 (s, 1H, H3), 7.63 (d, *J*= 6.9 Hz, 2H, H16, H20), 7.48 (m, 1H, H18), 7.43 (d, *J*= 6.9 Hz, 2H, H17, H19), 6.68 (d, *J*= 8.1 Hz, 1H, H7), 5.47 (dd, *J*= 5.4 Hz, *J*= 7.8 Hz, 1H, H6), 4.12 (d, *J*= 5.4 Hz, 1H, H5), 2.35–1.96 (m, 6H, H9, H10, H11); ^13^C NMR (CDCl3) δ(ppm): 181.6 (C1, C=O), 176.2 (C12, C=O), 137.2 (C3), 133.1 (C16, C20), 131.0 (C18), 128.8 (C17, C19), 122.8 (C4), 122.0 (C7), 119.5 (C15), 119.2 (q, *J*_C-F_= 321 Hz, C22, CF_3_), 111.8 (C6), 92.7 (C14), 84.7 (C13), 53.5 (C8), 36.4 (C5), 28.4 (C11), 28.0 (C9), 16.3 (C10); HRMS-DART^+^ (19 eV): elemental composition C_20_H_17_F_3_NO_5_S, [M+1]^+^, correlation (error of −0.17 ppm) exact value 440.0779 Daltons and precise value of 440.0778 Daltons; complementarily, the provided unsaturation data (12.5) were in agreement with the structure.

1-(3-(3-phenylpropioloyl)-1-((trifluoromethyl)sulfonyl)-1,4-dihydropyridin-4-yl)-cyclohexane-1-carboxylic acid (**3c**), white solid was obtained with 93% isolated yield (0.49 g); mp 140–142 °C; IR (cm^−1^): 2959 (COOH), 2198 (C≡C), 1678 (C=O), 1619 (C=C); ^1^H NMR (CDCl_3_) δ (ppm): 8.17 (s, 1H, H3), 7.62 (d, *J* = 9.0 Hz, 2H, H18, H22), 7.50–7.40 (m, 3H, H19, H20, H21), 6.77 (d, *J* = 6.9 Hz, 1H, H7), 5.53 (dd, *J* = 6.0 Hz, *J* = 7.8 Hz, 1H, H6), 4.02 (d, *J* = 5.7 Hz, 1H, H5), 2.08–2.05 (m, 2H, H9), 1.67–1.63 (m, 4H, H10, H13), 1.33–1.24 (m, 2H, H12), 1.20–1.05 (m, 2H, H11); ^13^C NMR (CDCl_3_) δ (ppm): 178.9 (C1, C=O), 176.0 (C14, C=O), 137.1 (C3), 133.1 (C18, C22), 131.0 (C20), 128.7 (C19, C21), 122.9 (C7), 122.5 (C4), 119.5 (C17), 119.2 (q, *J*_C-F_ = 321 Hz, C24, CF_3_), 111.6 (C6), 92.4 (C15), 84.6 (C16), 53.4 (C8), 38.6 (C5), 30.0 (C9), 29.7 (C13), 29.5 (C10), 25.4 (C12), 23.3 (C11); HRMS-DART^+^ (19 eV): elemental composition C_22_H_21_F_3_NO_5_S, [M+1]^+^, correlation (error of −2.12 ppm) exact value 468.1092 Daltons and precise value of 468.1082 Daltons; complementarily, the provided unsaturation data (12.5) were in agreement with the structure.

2-methyl-2-(3-(3-(4-(trifluoromethyl)phenyl)propioloyl)-1-((trifluoromethyl)sulfonyl)-1,4-dihydropyridin-4-yl)-propanoic acid (**3d**), red solid was obtained with 82% isolated yield (0.31g); mp 118–120 °C; IR (cm^−1^): 2985 (COO-H), 2200 (C≡C), 1680 (C=O), 1596 (C=C); ^1^H NMR (CDCl_3_) δ (ppm): 8.13 (s, 1H, H5), 7.72–7.65 (m, 4H, H15, H16, H18, H29), 6.73 (d, *J* = 7.5 Hz, 1H, H9), 5.47 (dd, *J* = 7.8 Hz, *J* = 5.4 Hz, 1H, H8), 4.20 (m, 1H, H7), 1.12 (s, 3H, H3), 1.10 (s, 3H, H10); ^13^C NMR (CDCl_3_) δ (ppm): 179.9 (C1, C=O), 175.9 (C11, C=O), 167.96 (C17), 138.0 (C5), 133.3 (C15, C19), 131.02 (C16, C18), 129.9 (d, *J*_C-F_ = 164 Hz, C27, CF_3_), 125.85 (C9), 122.9 (C6), 119.1 (q, *J*_C-F_ = 321 Hz, C21, CF_3_), 112.2 (C8), 90.1 (C12), 85.8 (C13), 63.7 (C2), 47.5 (C7), 22.1 (C3), 20.4 (C10); HRMS-DART^+^ (19 eV): elemental composition C_20_H_16_F_6_NO_5_S, [M+1]^+^, correlation (error of −1.01 ppm) exact value 496.0653 Daltons and precise value of 496.0648 Daltons; complementarily, the provided unsaturation data (11.5) were in agreement with the structure.

2-(3-(3-(4-methoxyphenyl)propioloyl)-1-((trifluoromethyl)sulfonyl)-1,4-dihydropyridin-4-yl)-2-methylpropanoic acid (**3e**), red solid was obtained with 76% isolated yield (0.27g); mp 146–148 °C; IR (cm^−1^): 2923 (COO-H), 2193 (C≡C), 1683 (C=O), 1600 (C=C); ^1^H NMR (CDCl_3_) δ (ppm): 9.48 (s, 1H, OH), 8.14 (s, 1H, H5), 7.54 (d, *J* = 9.0 Hz, 2H, H15, H19), 6.91 (d, *J* = 8.7 Hz, 2H, H16, H18), 6.75 (d, *J* = 5.4 Hz, 1H, H9), 5.48–5.46 (dd, *J* = 7.8 Hz, *J* = 5.7 Hz, 1H, H8), 4.13 (d, *J* = 5.7 Hz, 1H, H7), 1.24 (s, 3H, H3), 1.12 (s, 3H, H10); ^13^C NMR (CDCl_3_) δ (ppm): 181.4 (C1, C=O), 176.2 (C11, C=O), 162.0 (C17), 137.5 (C5), 135.2 (C15, C19), 122.9 (C9), 122.9 (C6), 119.1 (q, *J*_C-F_ = 321 Hz, C22, CF_3_), 114.6 (C16, C18), 112.3 (C8), 111.3 (C14), 93.8 (C12), 84.7 (C13), 55.5 (C2), 37.5 (C7), 22.7 (C3), 19.9 (C10); HRMS-DART^+^ (19 eV): elemental composition C_20_H_19_F_3_NO_5_S, [M+1]^+^, correlation (error of −3.57 ppm) exact value 458.0885 Daltons and precise value of 458.0868 Daltons; complementarily, the provided unsaturation data (11.5) were in agreement with the structure.

2-(3-(hept-2-ynoyl)-1-((trifluoromethyl)sulfonyl)-1,4-dihydropyridin-4-yl)-2-methylpropanoic acid (**3f**), yellow solid was obtained with 77% isolated yield (0.33 g); mp 128–130 °C; IR (cm^−1^): 2937 (COOH), 2200 (C≡C), 1686 (C=O), 1614 (C=C); ^1^H NMR (CDCl_3_) δ (ppm): 11.2 (s, 1H, OH), 7.94 (s, 1H, H5), 6.63 (d, *J* = 7.8 Hz, 1H, H9), 5.42 (dd, *J* = 5.4 Hz, *J* = 7.5 Hz, 1H, H8), 3.97 (d, *J* = 5.7 Hz, 1H, H7), 2.36 (t, 2H, H14), 1.57–1.47 (m, 2H, H15), 1.44–1.31 (m, 2H, H16), 1.01 (s, 3H, H3), 0.99 (s, 3H, H10), 0.85 (t, 3H, H17); ^13^C NMR (CDCl_3_) δ (ppm): 182.2 (C1, C=O), 176.2 (C11, C=O), 137.1 (C5), 122.7 (C9), 119.1 (q, *J*_C-F_ = 321 Hz, C19, CF_3_), 112.0 (C8) 95.8 (C13), 77.4 (C12), 47.6 (C2), 37.3 (C7), 22.3 (C3), 21.8 (C), 19.7 (C10), 18.5 (C14), 13.2 (C17); HRMS-DART^+^ (19 eV): elemental composition C_17_H_21_F_3_NO_5_S, [M+1]^+^, correlation (error of −2.97 ppm) exact value 408.1092 Daltons and precise value of 408.1080 Daltons; complementarily, the provided unsaturation data (7.5) were in agreement with the structure.

1-(3-(hept-2-ynoyl)-1-((trifluoromethyl)sulfonyl)-1,4-dihydropyridin-4-yl)-cyclobutane-1-carboxylic acid (**3g**), yellow solid was obtained with 72% isolated yield (0.32 g); mp 116–118 °C; IR (cm^−1^): 2930 (COOH), 2221 (C≡C), 1720 (C=O), 1632 (C=C); ^1^H NMR (CDCl_3_) δ (ppm): 7.95 (s, 1H, H3), 6.06 (s, 1H, H7), 4.71 (s, 1H, H6), 3.8 (s, 1H, H5), 2.49–2.36 (m, 6H, H9, H11, H15), 2.13–2.10 (m, 2H, H16), 1.64–1.59 (m, 2H, H27), 1.49–1.46 (m, 2H, H10), 0.95 (t, 3H, H18); ^13^C NMR (CDCl_3_) δ (ppm): 181.7 (C1, C=O), 176.4 (C12, C=O), 136.9 (C3), 122.8 (C4), 121.9 (C7), 119.8 (q, *J*_C-F_ = 321 Hz, C20, CF_3_), 111.7 (C6), 96.1 (C13), 77.4 (C12), 53.4 (C8), 37.8 (C5), 35.8 (C9), 29.5 (C11), 28.2 (C14), 28.0 (C16), 25.0 (C10), 21.8 (C17), 18.5 (C15), 13.2 (C18); HRMS-DART^+^ (19 eV): elemental composition C_18_H_21_F_3_NO_5_S, [M+1]^+^, correlation (error of +2.60 ppm) exact value 420.1092 Daltons and precise value of 420.1081 Daltons; complementarily, the provided unsaturation data (8.5) were in agreement with the structure.

1-(3-(hept-2-ynoyl)-1-((trifluoromethyl)sulfonyl)-1,4-dihydropyridin-4-yl)-cyclopentane-1-carboxylic acid (**3h**), yellow solid was obtained with 74% isolated yield (0.34 g); mp 126–128 °C; IR (cm^−1^): 2937 (COOH), 2224 (C≡C), 1682 (C=O), 1613 (C=C); ^1^H NMR (CDCl_3_) δ (ppm): 8.00 (s, 1H, H3), 6.66 (d, *J* = 7.8 Hz, 1H, H7), 5.50 (dd, *J* = 5.4 Hz, *J* = 7.8 Hz, 1H, H6), 4.12 (d, *J* = 5.4 Hz, 1H, H5), 2.46 (t, 2H, H16), 2.10–2.05 (m, 2H, H9), 1.63–1.28 (m, 12H, H10, H11, H12, H17, H18), 0.96 (t, 3H, H19); ^13^C NMR (CDCl_3_) δ (ppm): 181.6 (C1, C=O), 176.4 (C13, C=O), 137.1 (C3), 123.3 (C4), 122.2 (C7), 119.1 (q, *J*_C-F_ = 321 Hz, C21, CF_3_), 112.8 (C6), 96.4 (C15), 77.5 (C14), 60.5 (C8), 35.9 (C5), 33.1 (C9), 31.5 (C12), 29.6 (C17), 24.0 (C10), 23.9 (C11), 22.0 (C18), 18.7 (C16), 13.4 (C19); HRMS-DART^+^ (19 eV): elemental composition C_19_H_23_F_3_NO_5_S, [M+1]^+^, correlation (error of +1.85 ppm) exact value 434.1249 Daltons and precise value of 434.1257 Daltons; complementarily, the provided unsaturation data (8.5) were in agreement with the structure.

1-(3-(hept-2-ynoyl)-1-((trifluoromethyl)sulfonyl)-1,4-dihydropyridin-4-yl)-cyclohexane-1-carboxylic acid (**3i**), yellow solid was obtained with 87% isolated yield (0.41 g); mp 130–132 °C; IR (cm^−1^): 2938 (COOH), 2223 (C≡C), 1680 (C=O), 1613 (C=C); ^1^H NMR (CDCl_3_) δ (ppm): 8.01 (s, 1H, H3), 6.73 (d, *J* = 7.8 Hz, 1H, H7), 5.51 (dd, *J* = 5.7 Hz, *J* = 7.8 Hz, 1H, H6), 3.94 (d, *J* = 5.7 Hz, 1H, H5), 2.44 (t, 2H, H17), 2.04–2.00 (m, 2H, H9), 1.66–0.92 (m, 15H, H10-H13, H18-H20); ^13^C NMR (CDCl_3_) δ (ppm): 180.5 (C1, C=O), 176.3 (C14, C=O), 137.0 (C3), 122.8 (C7), 122.7 (C4), 119.5 (q, *J*_C-F_ = 321 Hz, C22, CF_3_), 111.7 (C6), 95.8 (C16), 77.4 (C15), 53.4 (C8), 38.6 (C5), 38.4 (C9), 29.9 (C13), 29.6 (C18), 25.4 (C10), 23.2 (C12), 21.9 (C11), 18.7 (C19), 18.6 (C17), 13.4 (C20); HRMS-DART^+^ (19 eV): elemental composition C_20_H_25_F_3_NO_5_S, [M+1]^+^, correlation (error of −1.92 ppm) exact value 448.1405 Daltons and precise value of 448.1396 Daltons; complementarily, the provided unsaturation data (8.5) were in agreement with the structure.

2-methyl-2-(3-(non-2-ynoyl)-1-((trifluoromethyl)sulfonyl)-1,4-dihydropyridin-4-yl)-propanoic acid (**3j**), yellow solid was obtained with 68% isolated yield (0.37g); mp 122–124 °C; IR (cm^−1^): 2930 (COO-H), 2223 (C≡C), 1678 (C=O), 1613 (C=C); ^1^H NMR (CDCl_3_) δ (ppm): 10.19 (s, 1H, OH), 7.79 (s, 1H, H5), 6.47 (d, *J* = 7.8 Hz, 1H, H9), 5.24 (dd, *J* = 5.7 Hz, *J* = 7.8 Hz, 1H, H8), 3.82 (d, *J* = 5.4 Hz, 1H, 7H), 2.20 (t, 2H, H14), 1.43–1.33 (m, 2H, H15), 1.22–1.02 (m, 3H, H3, H16-H18), 0.86 (s, 3H, H10); 0.85 (s, 3H, H19); ^13^C NMR (CDCl_3_) δ (ppm): 182.2 (C1, C=O), 176.3 (C11, C=O), 137.2 (C5), 122.8 (C9), 122.6 (C6) 119.3 (q, *J*_C-F_ = 321 Hz, C21 CF_3_), 112.1 (C8), 96.1 (C13), 77.4 (C12), 47.7 (C2), 37.3 (C7), 31.1 (C17), 28.5 (C16), 27.6 (C15), 22.5 (C3), 22.3 (C18), 19.8 (C10), 19.0 (C14), 13.9 (C19); HRMS-DART^+^ (19 eV): elemental composition C_19_H_25_F_3_NO_5_S, [M+1]^+^, correlation (error of −0.89 ppm) exact value 436.1405 Daltons and precise value of 436.1401 Daltons; complementarily, the provided unsaturation data (7.5) were in agreement with the structure.

1-(3-(non-2-ynoyl)-1-((trifluoromethyl)sulfonyl)-1,4-dihydropyridin-4-yl)-cyclobutane-1-carboxylic acid (**3k**), yellow solid was obtained with 60% isolated yield (0.37g); mp 122–124 °C; IR (cm^−1^): 2930 (COOH), 2216 (C≡C), 1697 (C=O), 1613 (C=C); ^1^H NMR (CDCl_3_) δ (ppm): 7.99 (s, 1H, H3), 6.64 (d, *J* = 8.1 Hz, 1H, H7), 5.49 (dd, *J* = 5.4 Hz, *J* = 7.8 Hz, 1H, H6), 4.10 (d, *J* = 5.4 Hz, 1H, H5), 2.44 (t, 2H, H20), 2.07–2.03 (m, 2H, H9), 1.64–1.57 (m, 4H, H10, H11), 1.48–1.41 (m, 4H, H16, H17), 1.33–1.25 (m, 4H, H18, H19), 0.89 (t, 3H, H20); ^13^C NMR (CDCl_3_) δ (ppm): 182.1 (C1, C=O), 176.2 (C12, C=O), 137.2 (C3), 122.8 (C7), 122.6 (C4) 119.2 (q, *J*_C-F_ = 321 Hz, C22, CF_3_), 112.1 (C6), 96.1 (C14), 77.4 (C13), 47.7 (C8), 37.3 (C5), 31.1 (C17), 28.5 (C9), 27.6 (C11), 22.5 (C16), 22.3 (C18), 19.8 (C19), 19.0 (C10), 18.3 (C15), 13.9 (C20); HRMS-DART^+^ (19 eV): elemental composition C_20_H_25_F_3_NO_5_S, [M+1]^+^, correlation (error of −3.64 ppm) exact value 448.1405 Daltons and precise value of 448.1389 Daltons; complementarily, the provided unsaturation data (8.5) were in agreement with the structure.

1-(3-(non-2-ynoyl)-1-((trifluoromethyl)sulfonyl)-1,4-dihydropyridin-4-yl)-cyclopentane-1-carboxylic acid (**3l**), yellow solid was obtained with 65% isolated yield (0.37g); mp 134–136 °C; IR (cm^−1^): 2927 (COOH), 2223 (C≡C), 1679 (C=O), 1614 (C=C); ^1^H NMR (CDCl_3_) δ (ppm): 10.49 (s, 1H, OH), 7.99 (s, 1H, H3), 6.64 (d, *J* = 8.1 Hz, 1H, H7), 5.49 (dd, *J* = 5.4 Hz, *J* = 7.8 Hz, 1H, H6), 4.10 (d, *J* = 5.4 Hz, 1H, H5), 2.44 (t, 2H, H16), 2.07–2.03 (m, 2H, H9), 1.64–1.57 (m, 4H, H12, H17), 1.48–1.41 (m, 4H, H18, H19), 1.33–1.25 (m, 6H, H10, H11, H20), 0.89 (t, 3H, H21); ^13^C NMR (CDCl_3_) δ (ppm): 181.9 (C1, C=O), 176.5 (C13, C=O), 137.1 (C3), 123.3 (C4), 122.1 (C7), 119.3 (q, *J*_C-F_ = 321 Hz, C23, CF_3_), 112.8 (C6), 96.1 (C15), 77.5 (C14), 60.6 (C8), 43.6 (C5), 35.9 (C9), 33.1 (C12), 31.5 (C19), 31.1 (C17), 29.9 (C18), 28.5 (C10), 27.5 (C11), 23.9 (C20), 19.8 (C16), 13.9 (C21); HRMS-DART^+^ (19 eV): elemental composition C_21_H_27_F_3_NO_5_S, [M+1]^+^, correlation (error of −3.46 ppm) exact value 462.1562 Daltons and precise value of 462.1546 Daltons; complementarily, the provided unsaturation data (8.5) were in agreement with the structure.

1-(3-(non-2-ynoyl)-1-((trifluoromethyl)sulfonyl)-1,4-dihydropyridin-4-yl)-cyclohexane-1-carboxylic acid (**3m**), yellow solid was obtained with 74% isolated yield (0.37g); mp 110–112 °C; IR (cm^−1^): 2932 (COOH), 2217 (C≡C), 1760 (C=O), 1626 (C=C); ^1^H NMR (CDCl_3_) δ (ppm): 7.99 (s, 1H, H3), 6.71 (d, *J* = 7.8 Hz, 1H, H7), 5.46 (dd, *J* = 5.7 Hz, *J* = 7.8 Hz, 1H, H6), 3.92 (d, *J* = 5.7 Hz, 1H, H5), 2.42 (t, 2H, H17), 2.02–1.98 (m, 2H, H9), 1.65–1.56 (m, 4H, H13, H10), 1.44–1.39 (m, 4H, H12, H18), 1.31–1.24 (m, 8H, H11, H19, H20, H21), 0.88 (t, 3H, H22); ^13^C NMR (CDCl_3_) δ (ppm): 180.0 (C1, C=O), 176.3 (C14, C=O), 137.0 (C3), 122.8 (C4), 122.5 (C7), 119.3 (q, *J*_C-F_ = 321, C24, CF_3_), 111.6 (C6), 96.0 (C16), 77.4 (C15), 53.3 (C8), 38.6 (C5), 38.4 (C20), 31.2 (C9), 29.9 (C13), 29.3 (C18), 28.6 (C19), 27.6 (C10), 25.4 (C12), 23.2 (C11), 22.4 (C21), 19.8 (C17), 13.9 (C22); HRMS-DART^+^ (19 eV): elemental composition C_22_H_29_F_3_NO_5_S, [M+1]^+^, correlation (error of −3.34 ppm) exact value 476.1718 Daltons and precise value of 476.1702 Daltons; complementarily, the provided unsaturation data (8.5) were in agreement with the structure.

2-(3-(3-cyclopentylpropioloyl)-1-((trifluoromethyl)sulfonyl)-1,4-dihydropyridin-4-yl)-2-methylpropanoic acid (**3n**), yellow oil was obtained with 74% isolated yield (0.31 g); IR (cm^−1^): 2939 (COOH), 2206 (C≡C), 1701 (C=O), 1613 (C=C); ^1^H NMR (CDCl_3_) δ (ppm): 10.02 (s, 1H, OH), 7.97 (s, 1H, H5), 6.66 (d, *J* = 7.8 Hz, 1H, H9), 5.43 (dd, *J* = 5.7 Hz, *J* = 7.8 Hz, 1H, H8), 3.99 (d, *J* = 5.4 Hz, 1H, H7), 2.80–2.78 (m, 1H, H14), 1.97–1.66 (m, 4H, H15, H18), 1.58–1.56 (m, 4H, H16, H17), 1.04 (s, 3H, H3), 1.02 (s, 3H, H10); ^13^C NMR (CDCl_3_) δ (ppm): 182.1 (C1, C=O), 176.6 (C11, C=O), 137.4 (C5), 123.0 (C9), 122.9 (C6), 119.3 (q, *J*_C-F_ = 321 Hz, C20, CF_3_), 112.1 (C8), 100.3 (C13), 94.5 (C12), 47.7 (C2), 37.4 (C7), 33.2 (C14), 33.2 (C15), 30.0 (C18), 25.2 (C16), 25.1 (C17), 22.4 (C3), 19.8 (C10); HRMS-DART^+^ (19 eV): elemental composition C_18_H_21_F_3_NO_5_S, [M+1]^+^, correlation (error of +4.13 ppm) exact value 420.1092 Daltons and precise value of 420.1109 Daltons; complementarily, the provided unsaturation data (8.5) were in agreement with the structure.

2-methyl-2-(3-(3-(thiophen-3-yl)propioloyl)-1-((trifluoromethyl)sulfonyl)-1,4-dihydropyridin-4-yl)-propanoic acid (**3o**), white solid was obtained with 72% isolated yield (0.29 g); mp 156–158 °C; IR (cm^−1^): 2927 (COOH), 2197 (C≡C), 1758 (C=O), 1623 (C=C); ^1^H NMR (CDCl_3_) δ (ppm): 8.12 (s, 1H, H5), 7.79 (d, *J* = 3.0 Hz, 1H, H18), 7.36 (dd, *J* = 3.0 Hz, *J* = 5.1 Hz, 1H, H17), 7.25 (s, 1H, H16) 6.76 (d, *J* = 7.8 Hz, 1H, 9H), 5.50 (dd, *J* = 6.0 Hz, *J* = 8.4 Hz, 1H, H8), 4.12 (d, *J* = 5.7 Hz, 1H, H7), 1.12 (s, 6H, H3, H10); ^13^C NMR (CDCl_3_) δ (ppm): 181.9 (C1, C=O), 176.2 (C11, C=O), 137.4 (C5), 134.3 (C18), 130.2 (C16), 126.5 (C17), 123.0 (C9), 122.7 (C6), 119.3 (q, *J*_C-F_ = 321 Hz, C20, CF_3_), 118.8 (C14), 112.2 (C8), 88.1 (C12), 85.0 (C13), 47.8 (C2), 37.5 (C7), 22.6 (C3), 19.8 (C10); HRMS-DART^+^ (19 eV): elemental composition C_17_H_15_F_3_NO_5_S_2_, [M+1]^+^, correlation (error of −2.71 ppm) exact value 434.0343 Daltons and precise value of 434.0332 Daltons; complementarily, the provided unsaturation data (11.5) were in agreement with the structure.

2-(3-(3-(ferrocenyl-1-yl)propioloyl)-1-((trifluoromethyl)sulfonyl)-1,4-dihydropyridin-4-yl)-propanoic acid (**3p**), red solid was obtained with 48% isolated yield (0.16 g); mp 148–150 °C; IR (cm^−1^): 2929 (COOH), 2188 (C≡C), 1725 (C=O), 1614 (C=C).;^1^H NMR (CDCl_3_) δ (ppm): 8.10 (s, 1H, H5), 6.75 (d, *J* = 8.1 Hz, 1H, H9), 5.50 (d, *J* = 6.3 Hz, 1H, H8), 4.62 (s, 2H, H16, H17, Fc), 4.41 (s, 2H, H15, H18, Fc), 4.27 (s, 5H, H19-H23, Fc) 4.12 (d, *J* = 5.7 Hz, 1H, H7), 1.13 (s, 6H, H3, H10); ^13^C NMR (CDCl_3_) δ (ppm): 181.1 (C1, C=O), 175.9 (C11, C=O), 136.7 (C5), 123.0 (C9), 119.3 (q, *J*_C-F_ = 321 Hz, C25, CF_3_), 112.2 (C8), 96.1 (C13), 83.0 (C12), 73.1 (C15, C18, Fc), 73.0 (C14, Fc), 70.5 (C19-C23, Fc), 59.6 (C2), 37.6 (C7), 29.7 (C16, C18, Fc), 22.7 (C3), 20.1 (C10); HRMS-DART^+^ (19 eV): elemental composition C_23_H_21_F_3_FeNO_5_S, [M+1]^+^, correlation (error of −1.43 ppm) exact value 536.0441 Daltons and precise value of 536.0434 Daltons; complementarily, the provided unsaturation data (14.0) were in agreement with the structure.

## 4. Conclusions

A series of 16 dihydropyridine carboxylic acids were synthesized through a process involving the activation of pyridine derivatives ynones with triflic anhydride, followed by the nucleophilic addition of *bis*(trimethylsilyl) ketene acetals; the obtained molecules were unequivocally characterized by common spectroscopic modes complemented through an X-ray diffraction analysis of one of the target molecules. In general, the obtained compounds showed interesting cytotoxic activity for HCT-15 cell line; it is worth noting that two of them, **3a**,**b**, displayed higher potency (7.94 ± 1.6 μM and 9.24 ± 0.9 μM, respectively). Consequently, it was considered important to perform theoretical calculations employing density functional theory (DFT) using the B3LYP/6–311++G(d,p). Interesting results were achieved: computational chemistry is a powerful tool used for the molecular property’s determination; it can simulate and explain how molecules interact at molecular level in the fields of chemistry, pharmacology, and drug design; additionally, complex chemical processes can also be simulated. Computational chemistry allows for precise calculations that yield highly accurate results. Moreover, this approach can eliminate the need for expensive experiments. Moreover, employing Osiris Property Explorer, it was evidenced that molecules **3a**,**b** have physicochemical characteristics that consider them to be orally active drugs; in addition, the performed docking studies showed a high affinity to PARP-1 protein. Finally, it is very convenient to highlight that through evaluating the synthesis, in silico studies, and cytotoxic activity of two of the target compounds, new suitable antineoplasic agent candidates are now available.

## Data Availability

The data reported in this study are available upon request to mirruv@yahoo.com.mx.

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
