# Peer review of "Synthesis, Cytotoxic Activity and In Silico Study of Novel Dihydropyridine Carboxylic Acids Derivatives"

_ijms, 2023, doi:10.3390/ijms242015414_

Round 1

Reviewer 1 Report

This paper deal with the synthesis, analysis, and biological evaluation of a series of 16 dihydropyridine carboxylic acids. In particular, two of them showed a good cytotoxic activity along with a potential good oral bioavailability, as displayed by the use of Osiris Property Explorer. In addition, the performed docking studies showed a high affinity to PARP-1 protein showing that this class of compounds could be a new series of antineoplasic agents.

For these reasons, I consider this manuscript suitable for publication in IJMS, after some improvements:

-          In lines 135 and 136 is stated that only four compounds showed an inhibitory percentage greater than 50% in the HCT-50 cell line, but I can find 9 compounds with Inhibition of the growth (%) higher than 50% for HCT-15. Is there an error?

-          In Table 6, what compound did you use as reference?

-      Since the compounds you have synthetized and analyzed are new, it is recommended to include the NMR spectra in the Supplementary Materials.

Reviewer 2 Report

The work carries adequate merits, while some issues need to be addressed for next step processiong.

1.     The aim of the research is not well reflected in the abstract section. Please revise.

2.     The docking score less than -10 is not theoretically accepted as prospective for therapeutics. Therefore, I recommend revising the whole manuscript based on this aspect.

3.     The introduction is well written, however; to prove the novelty of the research the inefficiency/drawbacks of existing anticancer drugs and inevitability of incorporating new drugs need to be addressed.

4.     In section 2.2. Cytotoxic assay, what are the bases of choosing the single dose 25 μM?

5.     According to Line 134-136, only few compounds are mentioned to show the inhibition >50%; however, I suggest rechecking and revising because I see some other compounds have the inhibition percentage more than 50%

6.     The statement in the lines 139-140 is wrong, because some other IC50 values are lower than Cisplatin (means higher effect). Needs to be revised.

7.     Line 147-148; Where is the ADMET data? What about the in silico toxicity data? Drug likeliness could be predicted from those data.

8.     Line 467, the access date to the link should be pronounced.

9.     Line 532, clearly specify the gel used for column chromatography

10.  Based on the comment 2, authors should revise the conclusion.

11.  Throughout the whole manuscript, there are so many wrong formats

12.  In vitro, in silico should be italic.

English language and textual formatting need to be improved

Reviewer 3 Report

The manuscript "Synthesis, cytotoxic activity and in silico study of novel dihydropyridine carboxylic acids derivatives" describes the synthesis, characterization, and biological evaluation of a series of novel dihydropyridine carboxylic acid derivatives. The compounds were synthesized via a straightforward method and evaluated for their cytotoxicity against several cancer cell lines. The most active compounds were further studied using computational methods.

The topic is interesting and the manuscript is generally well-written. The chemistry section clearly describes the synthesis and characterization of the new compounds. The biological results demonstrate moderate anticancer activity for some of the derivatives. The computational studies provide useful insights into the properties and predicted binding modes of the active compounds. Overall, I think this is a nice contribution to the literature on dihydropyridine-based anticancer agents. I have just a few suggestions to further improve the quality of the manuscript:

  1. The Introduction could be expanded slightly to provide more background on dihydropyridines, their biological activities, and significance as anticancer drug candidates.
  2. In the Results section, the NMR data for compound 3a is described in detail but this level of detail is not provided for the other compounds. Consider adding a supplemental table with concise NMR data for all final compounds or just referring to the general characterization methods used.
  3. The computational modeling studies are interesting but very brief. Please provide more details on the methods, models, and parameters used for the DFT, docking, etc.
  4. There are some minor grammatical errors throughout that should be corrected.
  5. The quality of Figures 2, 6 and 7 could be improved.

6         6. Unify the style of the references in the References Section and add DOI in the cases it is possible. And use the same reference and citation (follow MDPI’s guidelines) style in the main text.

The writing is clear and straightforward, allowing the key information to be conveyed effectively. The authors use appropriate scientific terminology and style.

The grammar and sentence structure are generally correct, with only minor errors. For example, in a couple places the verb tense shifts within a sentence (e.g. "were acquired" instead of "was acquired"). But overall the grammar does not impede understanding.

Some sentences are a bit long or awkwardly phrased, but the meaning can still be determined. Breaking these up into shorter sentences could improve readability. For example "Fine-tuning a procedure from previous research, in which an imidazole was employed as substrate [23], the dihydropyridine carboxylic acid 3a was produced (Scheme 1):" could be rephrased for clarity.

The vocabulary level seems appropriate for a scientific publication in this field. The language used is formal without being overly complicated.

There are a few typos, like missing spaces between words, but these appear to be minor proofreading oversights.

Reviewer 4 Report

Minor language errors or typos were pointed out in the review report.

Round 2

Reviewer 2 Report

Authors addressed the suggestion

Reviewer 3 Report

The authors have successfully answered the comments and suggestions proposed

Reviewer 4 Report

Issues have been addressed. Agreed to be published in IJMS.